# Practices and drivers of self-medication with antibiotics among undergraduate medical students in Eastern Uganda: A cross-sectional study

Gloria Nakato[1], Pamella R. Adongo[1], Jacob Stanley Iramiot[2], Joshua Epuitai[1]*

1 Faculty of Health Sciences, Department of Nursing, Busitema University, Mbale, Uganda, 2 Faculty of Health Sciences, Department of Microbiology and Immunology, Busitema University, Mbale, Uganda

* joshuaepuitai@gmail.com

**Data Availability Statement:** All relevant data are within the paper and its Supporting Information files.

## Abstract

Self-medication with antibiotics remains one of the major drivers of antimicrobial resistance. Practices of self-medication among nursing and medical students have not been explored in our setting. This study sought to determine the prevalence and factors associated with self-medication with antibiotics among undergraduate university students pursuing health-related courses in Eastern Uganda. A descriptive cross-sectional study design was used. The study was done among undergraduate students who were doing undergraduate programs in Nursing, Anesthesia, and medicine at Busitema University. A self-administered questionnaire was used to collect data from 326 participants. Descriptive statistics were used in data analysis. The prevalence of self-medication with antibiotics was 93.8% (n = 300) of which 80% were either currently using self-medication or had self-medicated in the past six months. The common reasons for self-medication were the perception that it was a minor illness (55%), previous use of antibiotic (52%), a perception that they were health workers (50%), and the notion that they knew the right antibiotic for their condition (44%). Metronidazole (64%) and amoxicillin (65%) were the most commonly used antibiotics. Self-medication was most common for conditions such as peptic ulcer, diarrhea, and wound infections. Inappropriate drug use was common among participants on self-medication which occurred in the form of multiple use of antibiotics (64.4%, n = 194) and a tendency to switch to other antibiotics (58.5%) mostly because the former antibiotic was perceived not to be an effective treatment. The prevalence of self-medication with antibiotics was high among medical students. Prior use of the antibiotic and having a minor illness were the most common drivers of self-medication. Public health strategies should address the high misuse of antibiotics among medical students to negate the likely consequence of antimicrobial resistance.

## Background

Self-medication is one of the most common drivers of antimicrobial resistance the world over [1]. Overall, 50% of total antibiotics used are purchased over-the-counter without prescription

**Funding:** Research reported in this publication was supported by the Fogarty International Center of the National Institutes of Health, U.S. Department of State's Office of the U.S. Global AIDS Coordinator and Health Diplomacy (S/GAC), and President's Emergency Plan for AIDS Relief (PEPFAR) under Award Number IR25TW011213. The funders had no role in study design, data collection and analysis, decision to publish, or preparation of the manuscript. The content is solely the responsibility of the authors and does not necessarily represent the official views of the National Institutes of Health.

**Competing interests:** The authors have declared that no competing interests exist.

[2, 3]. Studies have revealed the burden of self-medication with antibiotics (SMA) to be higher in developing countries than in developed countries [4, 5]. In 2021, SMA was 3–19% in developing countries, while the prevalence was high ranging from 24% to 73.9% in African Countries, 36.1–45.8% in the Middle East, 29% in South America, and 4–75% in Asia [5]. In Uganda, SMA among university students was 63% [6] and 76% among residents in the rural setting [7]. In Northern Uganda, the commonly used antibiotics for self-medication were amoxicillin, metronidazole, and co-trimoxazole [7].

Studies have underscored several reasons for self-medication practices [4, 5]. Lack of medications in medical facilities, long waiting times, long distance to medical facilities, inability to pay for medical expenses, and flexibility to select desired medications were among the often cited reasons for self-medication [8]. Other reasons for self-medication were a lack of medical professionals, low quality of healthcare facilities, unregulated distribution of medicines, and patients' misconceptions about physicians which were more in developing countries [8].

Although self-medication may be beneficial when used judiciously, it is more frequently utilized incorrectly, without the right advice and justification [9]. In Jordan, 67.1% of adults wrongly thought that antibiotics could treat colds and coughs [9]. Inappropriate medication administration not only wastes resources but also puts users at risk for significant and even fatal side effects, misdiagnosis risk, drug interactions, and healthcare expenditure [10, 11]. Furthermore, inappropriate drug use increases the risk of antimicrobial resistance leading to increased cost of care, prolonged hospital stays, and poor health outcomes including mortality from conditions that were previously treatable with the common antibiotics [11, 12]. Despite the consequences of inappropriate use of antibiotics, the prevalence and factors associated with self-medication in Uganda are not well understood. The aim of conducting this study was to assess the practices and drivers of self-medication with antibiotics among university students at Busitema University, Mbale Campus in Eastern Uganda.

## Methods and materials

### Study design

This was a descriptive cross-sectional survey. The study design was suited for determining the prevalence and factors associated with self-medication with antibiotics among university students at Busitema University, Mbale Campus in Eastern Uganda.

### Study setting

This study was conducted at Busitema University at the Faculty of Health Sciences. The Faculty of Health Sciences is located in Mbale City within the premises of Mbale Regional Referral Hospital. The Faculty of Health Sciences, Mbale campus has an estimated 451 students pursuing undergraduate courses such as Bachelor of Nursing Science (BNS), Bachelor of Medicine and Bachelor of Surgery (MB ChB), and Bachelor of Science in Anesthesia (BNA). The University also offers post-graduate programs such as a Master's in Public Health and a Master of Medicine (internal medicine, pediatrics, obstetrics and gynecology).

### Sampling method and procedure

The study used a consecutive sampling technique to obtain participants of the study. This was done by recruiting participants who were available on the University premises at the time of data collection and were readily accessible to the researcher. All students in the Faculty of Health Sciences Busitema University with either university identity cards or student bio-data as proof of being Busitema University students were sampled. A list of undergraduate students

was obtained from the office of the academic registrar for verification. Students who met the inclusion criteria were invited to participate. Those who agreed were seen by the researcher who explained to them the nature of the study and assessed eligibility for the study. Students who were willing were consented and subsequently interviewed.

The study sample size of 326 was arrived at using the Kish-Leslie formula. The percentage population of nursing, medicine, and anesthesia students was used to calculate the proportionate number that each program would contribute to the total sample size.

### Study variables

Self-medication was taken as ever used antibiotics without a prescription from a qualified health worker. If yes, participants were asked if they had self-medicated in the past year, 6 months, 3 months, 1 month, or were currently on self-medication. Additional self-medication practices were assessed including the type of antibiotic that was used, the condition for which it was used, whether another antibiotic was concurrently used if they switched to another antibiotic and for what reason, and attitudes (whether self-medication was deemed to be an acceptable practice). Socio-demographic factors such as age in complete years, religion, source of income, marital status, year of study, access to health services, and presence of chronic illnesses were assessed. Health system-related factors such as lack of access to health care facilities, non-affordability of health care services, lack of drugs in health care facilities, lack of health workers in health care facilities, poor attitude of health workers in health care facilities towards patients, and lack of enforcement of antibiotic policies.

### Data collection tool and procedure

Data collection was conducted starting from 28[th] October 2022 to 30[th] November 2022. A self-administered semi-structured questionnaire written in English with both open-ended and closed-ended questions was used. The questionnaire was developed from a systematic literature review of several studies [4, 5]. The questionnaire contained two main parts: sociodemographic characteristics of the respondents and the practice of self-medication with antibiotics. Data was collected using the Kobo collect tool. The anonymized Kobo Collect tool minimized any potential bias, especially social desirability bias. Participants found within the university premises were approached and asked to fill in the online version of the questionnaire.

### Data processing and analysis

Data were processed (S1 Data) and analyzed using SPSS version 20. Descriptive statistics such as frequencies and percentages were used for categorical variables such as the socio-demographic data. Continuous variables such as age were summarized as means and standard deviations. The prevalence of self-medication with antibiotics was determined by dividing the number of students who had ever self-prescribed themselves with antibiotics by the total number of those recruited.

### Ethics clearance

Ethical approval was sought from the Mbale Regional Referral Hospital Research and Ethics Committee (Reference number: MRRHREC-OUT-011/2020). A written informed consent was obtained from participants and participation was voluntary. The privacy of participants was maintained at all times.

## Results

### Sociodemographic characteristics of the respondents

A total of 326 students at Busitema University Faculty of Health Sciences participated in the study (Table 1). The average age of the respondents was 25.7 (±4.9) years. The majority of the respondents were doing medicine and surgery 63.8% (n = 208).

### Prevalence of self-medication with antibiotics

Although a majority of the respondents 235 (72.1%) reported that self-medication is not an acceptable practice, the prevalence of self-medication with antibiotics was high (93.8%) (Table 2). Amoxicillin (65%) and metronidazole (64.1%) were the most common self-pre-scribed antibiotics. Diarrhea (51.3%), peptic ulcers (35.3%), and painful wounds (32.4%) were the most common conditions where antibiotics were mostly used as self-medication. More than half 170 (52%) of the participants reported that they only sought professional prescrip-tions when the condition became severe, while some sought professional help following no improvement 103 (31.6%) and adverse reactions 32 (9.2%) The reasons for self-medication

**Table 1. Description of socio-demographic characteristics of the respondents.**

| Variable | Frequency n = 326(%) | Mean | SD |
|---|---|---|---|
| **Age (years) continuous data** | | 25.7 | 4.9 |
| **Gender** | | | |
| Female | 169 (51.8) | | |
| Male | 157 (48.2) | | |
| **Marital status** | | | |
| Single | 272 (83.5) | | |
| Married | 54 (16.5) | | |
| **Employment status** | | | |
| Employed | 65 (23) | | |
| Unemployed | 251 (77) | | |
| **Religion** | | | |
| Christian | 300 (92) | | |
| Moslem | 26 (8) | | |
| **Residence** | | | |
| Rural | 102 (31.3) | | |
| Urban | 224 (68.7) | | |
| **Year of study** | | | |
| Pre-clinical | 183 (56.1) | | |
| Clinical | 143 (43.9) | | |
| **Program of study** | | | |
| MB.Ch.B | 208 (63.8) | | |
| BNA | 62 (19) | | |
| BNS | 56 (17.2) | | |
| **Annual frequency of medical check-ups** | | | |
| Never | 173 (53.1) | | |
| One and above | 153 (46.9) | | |
| **Mode of university admission** | | | |
| High school entry scheme | 247(75.8) | | |
| Diploma entry scheme | 79(24.2) | | |

**Table 2. Prevalence and practices of self-medication.**

| Variable | Frequency N = 306 (%) |
|---|---|
| **Prevalence of self-medication** | |
| Yes | 306(93.8) |
| Never | 20(6.2) |
| **Last time of self-medication** | |
| Currently self-medicating | 63(20.6) |
| Past 1 month | 81(26.5) |
| Past 3 months | 56 (18.3) |
| Past 6 months | 46 (15) |
| Past 1 year | 60 (19.6) |
| **Source of antibiotics** | |
| Private drug shop*, medical clinic, or roadside selling point[a] | 250 (81.7) |
| Pharmacy | 160 (52.3) |
| A friend | 39 (12.8) |
| Leftovers from previous use | 33 (10.8) |
| **What did you consider when buying the antibiotic?** | |
| Type of antibiotics | 241 (78.8) |
| Price of antibiotics | 159 (52) |
| Adverse reactions | 44 (14.4) |
| Brand of antibiotics | 34 (11.1) |
| **Self-medicated antibiotics** | |
| Amoxicillin | 199(65) |
| Amoxicillin/Cloxacillin | 49(16) |
| Metronidazole | 196(64.1) |
| Co-trimoxazole | 105(34.3) |
| Ciprofloxacin | 88(28.9) |
| Erythromycin | 46(15) |
| Ceftriaxone and Cefixime | 80(26.1) |
| Doxycycline, Tetracycline & Chloramphenicol | 55(18) |
| **Health conditions for self-medication** | |
| Diarrhea | 157(51.3) |
| Peptic ulcers | 108(35.3) |
| Painful wounds | 99(32) |
| Fever | 83(27) |
| Muscle pain | 78(25.5) |
| Cough and throat pain | 63(20.6) |
| **Acceptance of antibiotic self-medication** | |
| Not acceptable practice | 235(72.1) |
| Acceptable/good practice | 91(27.9) |

Drug shop* = Class C over the counter drug outlets

Roadside selling point[a] = drugs sold openly on the roadside & streets

were mostly attributed to the perception that the illness was minor 180 (55.2%), prior use of the drug 171 (52.5%), and the perception that they were health workers 163 (50%). The other reasons given for self-medication included perceived knowledge of the right antibiotic for their condition 144 (44.2%), lack of services in the hospital 108 (33.1%), high costs of consultation 84 (25.8%) and laboratory fees 56 (17.2%), SMA saves time 95 (29.1%), friend's advice 81 (24.9%), and mistreatment while in the hospital 38 (11.7%).

**Table 3. Description of SMA practices.**

| Variable | Frequency N = 306 (%) |
|---|---|
| **Switched antibiotics** | |
| No | 214 (69.9) |
| Yes | 94 (30.7) |
| **Reasons for switching antibiotics (n = 94)** | |
| The former antibiotics did not work | 55 (58.5) |
| To reduce adverse reactions | 14 (14.9) |
| The latter one was cheaper | 13 (13.8) |
| The former antibiotics ran out | 12 (12.8) |
| **Multiple use of antibiotics** | |
| Yes | 197 (64.4) |
| No | 109 (35.6) |
| **Frequency of self-medication** | |
| Always when sick | 138 (45.1) |
| Sometimes when sick | 108 (35.3) |
| Rarely when sick | 60 (19.6) |

## Description of SMA practices

Among the participants who had self-medicated, 30.7% of them had ever switched antibiotics (Table 3). The perception that the antibiotic was not working was the most (58.5%) common reason given for switching to another antibiotic. More than half of students who had ever self-medicated (64.4%) used multiple classes of antibiotics.

## Discussion

The study sought to determine the prevalence and reasons for SMA among undergraduate students pursuing health programs. Although self-medication was deemed unacceptable, nearly all the respondents had used SMA. The common reasons for SMA were prior use of the antibiotic drug, a perception that the illness was minor, and notions that they were healthcare providers. The commonly used SMA was metronidazole and amoxicillin which were used for conditions such as diarrhea, peptic ulcer, and wound infections. Practices of inappropriate use of SMA were prevalent including multiple antibiotic use, switching to another antibiotic, and frequent SMA whenever they were sick.

Our study findings underscore a highly prevalent use of SMA, a finding which is consistent with the high-end range of 8–93% reported in a systematic review of 15 studies from Low- and Middle-income countries [13, 14]. In Africa, the median prevalence of SMA was 56% (interquartile range: 41–75%) which underscores a significantly higher prevalent use of SMA in our study [14]. Although the high prevalence of SMA in our study could be attributed to the extended period in which SMA was considered as ever self-medicated, a significant number of participants were currently using or had self-medicated in the past six months before the study. Similar findings have shown a higher prevalent use of SMA among students pursuing health programs than the general population which may likely negatively lead to poor prescription practices in the future among these groups of students upon qualification [10]. The study was conducted during the COVID-19 pandemic which might have fueled high practices of SMA especially in a low-income setting that was already crumbling with a high burden of infectious disease [15, 16]. The high SMA could be attributable to sub-optimal compliance with prescription regulations and a lack of enforcement of regulations that underpin

operations in drug outlets, clinics, and private pharmacies [17]. Our study findings note that drug outlets, which often do not conform to prescription regulations, were the main source of antibiotics for self-medication. Consistent with our study findings, SMA may be due to a poor healthcare system which is often marked by nonchalant healthcare providers, and inaccessible & unaffordable services [18].

Previous studies have cited several reasons underlying SMA practices [6, 7, 14]. Consistent with our study findings, SMA is often driven by a desire for quick relief, lack of awareness of the severity of the disease and need to seek healthcare, past good experience with the drug, and suggestions from relatives and friends [10, 14]. In our study, SMA was partly a result of perceptions among students that they were healthcare providers which underlines the need for self-care as well as the perception of being knowledgeable to self-diagnose and prescribe the required antibiotic [10]. Although the study did not explore the COVID-19-related factors for SMA, studies have identified fear of COVID-19 infection, restrictive measures of quarantine and isolation, psychological stress, and perceptions that SMA is time-saving as the major reasons for SMA during COVID-19 infection [15].

In our study, amoxicillin, metronidazole, and co-trimoxazole were the most commonly self-prescribed antibiotics, a finding that is consistent with the previous studies [13, 14, 19]. Besides the affordability, safety profile, and accessibility of these antibiotics [14], the wide use of these antibiotics in self-medication is a reflection of the commonly prescribed drugs in Uganda for treating most conditions [6, 7]. Widespread prescription practices of these particular drugs may have inadvertently led to participants getting experience using the drug, leftover, and hoarding the antibiotics [14]. Routine medical prescription of a particular antibiotic may have insinuated a practice of self-medication as patients learn to attribute certain signs or symptoms to the use of a particular antibiotic [14]. Similar to the commonly used antibiotics for self-medication in our study, diarrhea, peptic ulcers, and painful wounds were the most common conditions where participants resorted to self-medication [7]. However, it remains unclear whether the right medication, the right dose, the right condition, and the right duration of therapy were adhered to by the participants [20]. Studies from Southeast Asia cited that common cold, flu and cough were the conditions antibiotics were used as self-medication [18].

SMA often leads to inappropriate antibiotic use [13]. Irrational antibiotic use occurs in the form of failure to complete a full course of treatment upon the disappearance of symptoms, multiple uses of antibiotics, sharing of antibiotics, hoarding of drugs for future use, and switching to another antibiotic [10]. Our study findings concur with previous studies which note the prevalent inappropriate antibiotic use among participants who resorted to self-medication [10]. The negative implication of irrational drug use in promoting antimicrobial resistance will in the long run negate concerted public health efforts to control the emerging global epidemic of antimicrobial resistance [10]. Furthermore, multiple antibiotic use without a prescription from a physician may result in adverse drug interactions and adverse reactions [20].

The study should be interpreted in light of its limitations. Data collection was conducted by the first author who was also a student in the institution, and may have been familiar to some of the study participants. The familiarity may have led to social desirability bias. The study being cross-sectional is limited in its ability to infer causation. Therefore, we report the likely drivers or reasons for SMA.

## Conclusion

The practice of self-medication with antibiotics was high in our setting. The reasons for self-medication emanated from individual beliefs and attitudes regarding illness and healthcare

seeking behaviors. The high practices of SMA inadvertently led to inappropriate antibiotic use including multiple uses of antibiotics, the tendency to use another antibiotic when the former was perceived to be ineffective, and the practice of SMA whenever they were sick. Metronidazole and amoxicillin were the most commonly used drugs for SMA for mostly diarrhea, peptic ulcers, and painful wounds. The study findings have important public health implications which if not addressed may lead to antimicrobial resistance, drug-to-drug interactions, adverse drug reactions, and wastage of healthcare resources.

## Supporting information

**S1 Checklist. STROBE statement—checklist of items that should be included in reports of observational studies.**
(DOCX)

**S1 Data. Dataset on practices and drivers of SMA.**
(XLSX)

## Acknowledgments

We thank the faculty in the Department of Nursing, Busitema University for the mentorship and guidance during the study as well as the participants in this study for their contribution.

## Author Contributions

**Conceptualization:** Gloria Nakato.

**Data curation:** Gloria Nakato, Pamella R. Adongo, Jacob Stanley Iramiot, Joshua Epuitai.

**Formal analysis:** Gloria Nakato, Pamella R. Adongo, Jacob Stanley Iramiot, Joshua Epuitai.

**Funding acquisition:** Gloria Nakato, Pamella R. Adongo, Jacob Stanley Iramiot, Joshua Epuitai.

**Investigation:** Gloria Nakato, Pamella R. Adongo, Jacob Stanley Iramiot, Joshua Epuitai.

**Methodology:** Gloria Nakato, Pamella R. Adongo, Jacob Stanley Iramiot, Joshua Epuitai.

**Project administration:** Gloria Nakato, Pamella R. Adongo, Jacob Stanley Iramiot, Joshua Epuitai.

**Resources:** Gloria Nakato, Pamella R. Adongo, Jacob Stanley Iramiot, Joshua Epuitai.

**Software:** Gloria Nakato, Pamella R. Adongo, Jacob Stanley Iramiot, Joshua Epuitai.

**Supervision:** Jacob Stanley Iramiot, Joshua Epuitai.

**Validation:** Gloria Nakato, Pamella R. Adongo, Jacob Stanley Iramiot, Joshua Epuitai.

**Visualization:** Gloria Nakato, Pamella R. Adongo, Jacob Stanley Iramiot, Joshua Epuitai.

**Writing – original draft:** Gloria Nakato.

**Writing – review & editing:** Pamella R. Adongo, Jacob Stanley Iramiot, Joshua Epuitai.

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
