## [Decision Letter · Decision Letter 0]

27 Sep 2023

PONE-D-23-21270Practices and Drivers of Self-Medication with Antibiotics among Undergraduate Medical Students in Eastern Uganda: A Cross-sectional StudyPLOS ONE

Dear Dr. Epuitai,

Thank you for submitting your manuscript to PLOS ONE. After careful consideration, we feel that it has merit but does not fully meet PLOS ONE’s publication criteria as it currently stands. Therefore, we invite you to submit a revised version of the manuscript that addresses the points raised during the review process.

We look forward to receiving your revised manuscript.

Kind regards,

Joseph Olusesan Fadare

Academic Editor

PLOS ONE

“Research reported in this publication was supported by the Fogarty International Center of the National Institutes of Health, U.S. Department of State's Office of the U.S. Global AIDS Coordinator and Health Diplomacy (S/GAC), and President's Emergency Plan for AIDS Relief (PEPFAR) under Award Number IR25TW011213. The content is solely the responsibility of the authors and does not necessarily represent the official views of the National Institutes of Health.”

Reviewers' comments:

Reviewer's Responses to Questions

**Comments to the Author**

1. Is the manuscript technically sound, and do the data support the conclusions?

Reviewer #1: Yes

Reviewer #2: Yes

2. Has the statistical analysis been performed appropriately and rigorously? 

Reviewer #1: Yes

Reviewer #2: Yes

3. Have the authors made all data underlying the findings in their manuscript fully available?

Reviewer #1: Yes

Reviewer #2: Yes

4. Is the manuscript presented in an intelligible fashion and written in standard English?

Reviewer #1: Yes

Reviewer #2: Yes

5. Review Comments to the Author

Reviewer #1: The authors have produced an useful piece of work on the the prevalence and factors associated with self-medication in Uganda.

Abstract

1. The following statement was mentioned twice in the abstract:

"The major drivers for self-medication were having a minor illness, prior experience with the drug, and the perception that they were already healthcare workers"

Background

1. Generic name need to be used for medications.

Reviewer #2: Specific comments

Page 3, Line 51: Ampiclox, Septrin: It is preferable to write the medications by their internationally recognized names or to use the trademark sign as superscript i.e. Septrin ®

Page 8, Table 1: Distance to health facility? - This question may be relevant if the study is community-based or the university is non-residential. If the university has a health facility, you may want to re-consider inclusion of the question.

Page 8, Table 1: Question – Are you a direct student? - For the sake of readers not from your region, kindly explain what is meant by direct and upgrading student

Table 2 is too extensive (almost two pages): The authors are advised to revise it either by changing part of its contents to charts and including some as statements in the text.

Page 9, Table 2: Question - Last time you self-treated? - For the sake of consistency, the authors are advised to stick to the term "self-medication"

Page 9, Table 2: Source of antibiotics Drug shop/clinc: This may confuse the readers - drug shop/clinic.

The authors may need to explain the difference using foot notes.

Page 10, Table 2: At home - I will suggest using leftovers from previous use" instead of "at home

Page 10, Table 2: Street: What is meant by this? Kindly clarify

Page 11, Table 2: Acceptable practice, good practice: How does one differentiate between "acceptable" and "good " practice?

Page 13, Line 155 : nearly all the respondents had ever used SMA: Please delete "ever". It gives a contrary meaning to the sentence.

Page 15, Line 215: Conclusion: Please summarize. There is no need to repeat most of the study findings here

6. PLOS authors have the option to publish the peer review history of their article (what does this mean?). If published, this will include your full peer review and any attached files.

Reviewer #1: No

Reviewer #2: **Yes: **Prof. Joseph Fadare

---

## [Author Response · Author response to Decision Letter 0]

5 Oct 2023

We would like to extend our sincere gratitude for the helpful peer-review of our work as well as the privilege to re-submit a revised manuscript. We are happy to note that we have addressed the concerns as per the attached response to the reviewer’s comments. We have attached the manuscript with track changes to show the corresponding changes that were made on the manuscript in light of the reviewer’s comments. The manuscript has been revised as per the PLOS ONE style formats, while the ethics statement has also been included in section of methods and materials. We have revised the reference list and removed references which were outdated, repeated and or were not published in peer-reviewed journals. 

We thank you for the suggestion to include the role of the funders in the study. The funder’s statement now reads as: Research reported in this publication was supported by the Fogarty International Center of the National Institutes of Health, U.S. Department of State's Office of the U.S. Global AIDS Coordinator and Health Diplomacy (S/GAC), and President's Emergency Plan for AIDS Relief (PEPFAR) under Award Number IR25TW011213. The funders had no role in study design, data collection and analysis, decision to publish, or preparation of the manuscript. The content is solely the responsibility of the authors and does not necessarily represent the official views of the National Institutes of Health.

Comment Response Page 

Reviewer #1: The authors have produced an useful piece of work on the the prevalence and factors associated with self-medication in Uganda.

 We thank you for the positive appreciation of our work. 

Abstract

1. The following statement was mentioned twice in the abstract:

"The major drivers for self-medication were having a minor illness, prior experience with the drug, and the perception that they were already healthcare workers"

 We thank you for the insightful observation regarding repeated statements in the abstract. We have taken note of the comment and revised it as per the guidance Line 32-33 

Page 2

Background

1. Generic name need to be used for medications. We thank you for the helpful suggestion to use generic names for medications. We have taken note of the comment and revised it accordingly. As per the guidance, generic names have now been used in the work. Line 58-59

Page 2

Comment Response Page 

Page 3, Line 51: Ampiclox, Septrin: It is preferable to write the medications by their internationally recognized names or to use the trademark sign as superscript i.e. Septrin ® We thank you for pointing out this. We agree that medicines are written using their generic names. We have taken note of the comment and revised the naming of drugs to include generic names. Page 3, line 58-59

Page 8, Table 1: Distance to health facility? - This question may be relevant if the study is community-based or the university is non-residential. If the university has a health facility, you may want to re-consider inclusion of the question. We thank you for the insightful observation. Although the university is non-residential, all students have access to the teaching hospital which situated within the residential location for most participants. We have considered the guidance to leave out this variable as it may not be relevant for this study population. Page 8 table 1

Page 8, Table 1: Question – Are you a direct student? - For the sake of readers not from your region, kindly explain what is meant by direct and upgrading student

 We thank you for noticing the need to clarify this variable. We have taken note of the comment and revised this variable to mean the mode of university admission (diploma scheme or through high school) Page 8 

Table 1

Table 2 is too extensive (almost two pages): The authors are advised to revise it either by changing part of its contents to charts and including some as statements in the text.

 Thank you for the insightful observation. We agree table 2 was long. We have considered the guidance to include some statements as text which now has reduced the content of table 2 Table 2 

Page 10-11

Page 9, Table 2: Question - Last time you self-treated? - For the sake of consistency, the authors are advised to stick to the term "self-medication"

 We thank you for the comment to maintain consistency in the work. We have taken note of the comment and revised it accordingly. It is now captured as last time of self-medication Table 2

Page 9, Table 2: Source of antibiotics Drug shop/clinc: This may confuse the readers - drug shop/clinic. The authors may need to explain the difference using foot notes.

 We thank you for the comment. We have modified the naming to make it more meaningful for the readers. As guided, we have added the footnotes that describe what a drug shop is in our setting. We have added drugs sold in the streets to the section of drug shop/clinic since operate in almost the way Page 10 

Table 2

Page 10, Table 2: At home - I will suggest using leftovers from previous use" instead of "at home We thank you for the suggestion to use the term leftovers from the previous use. We have revised it to ‘leftovers from previous use’ instead of ‘at home’ Table 2

Page 10, Table 2: Street: What is meant by this? Kindly clarify We thank you for the comment. We have modified the variable to ‘roadside selling point’ and added the footnotes to describe what that means Page 10 

Table 2

Page 11, Table 2: Acceptable practice, good practice: How does one differentiate between "acceptable" and "good " practice? We thank you for the comment. We agree that it may be difficult to differentiate between ‘acceptable’ and ‘good’ practice. Because participants may have interpreted the two concepts synonymously, we have decided to combine the two. The variable is now worded as ‘acceptable/good/’ practice Page 11 

Table 2

Page 13, Line 155 : nearly all the respondents had ever used SMA: Please delete "ever". It gives a contrary meaning to the sentence.

 Thank you for the insightful observation. We agree the modifier gives a different meaning to it. We have revised the statement to make it clearer Line 324 

Page 13

Page 15, Line 215: Conclusion: Please summarize. There is no need to repeat most of the study findings here

 We thank you for the suggestion to summarize the conclusion. We have revised it as per the guidance. Page 15 

Line 391-400

---

## [Editor Report · Decision Letter 1]

8 Oct 2023

PONE-D-23-21270R1Practices and Drivers of Self-Medication with Antibiotics among Undergraduate Medical Students in Eastern Uganda: A Cross-sectional StudyPLOS ONE

Dear Dr. Epuitai,

Thank you for submitting your manuscript to PLOS ONE. After careful consideration, we feel that it has merit but does not fully meet PLOS ONE’s publication criteria as it currently stands. Therefore, we invite you to submit a revised version of the manuscript that addresses the points raised during the review process.

Page 17, Lines 289-90: Please check the referencing - Organization WH - To be written as WHO (in full) Page 17: References nos 2-10 - Please check these references; names of journals are missing. This sometimes happens with the software being used for referencing. Page 18, Reference no 20 - Same comment as above

We look forward to receiving your revised manuscript.

Kind regards,

Joseph Olusesan Fadare

Academic Editor

PLOS ONE

Journal Requirements:

Additional Editor Comments:

Thank you for implementing some of the points raised during the initial review in the updated manuscript. However, there are still some issues to be addressed before this manuscript can make progress. Please find below some of the issues to be addressed.

Page 17, Lines 289-90: Please check the referencing - Organization WH - To be written as WHO (in full)

Pages 17: References nos 2-10 - Please check these references; names of journals are missing. This sometimes happens with the software being used for referencing.

Page 18, Reference no 20 - Same comment as above

---

## [Author Response · Author response to Decision Letter 1]

9 Oct 2023

We thank you for the guidance to revise the reference list. We have taken note of it and revised the reference list according. The reference in line 289 has now been corrected from Organization WH to World Health Organization (WHO). The names of journals have now been included in all the references where the names of the journals were missing (Line 289-342). We thank you once again for the insightful comments that have now resulted in an improved structure of the manuscript.

---

## [Editor Report · Decision Letter 2]

10 Oct 2023

PONE-D-23-21270R2Practices and Drivers of Self-Medication with Antibiotics among Undergraduate Medical Students in Eastern Uganda: A Cross-sectional StudyPLOS ONE

Dear Dr. Epuitai,

Thank you for submitting your manuscript to PLOS ONE. After careful consideration, we feel that it has merit but does not fully meet PLOS ONE’s publication criteria as it currently stands. Therefore, we invite you to submit a revised version of the manuscript that addresses the points raised during the review process.

Thank you for updating the manuscript.

However, there are some outstanding issues. This may be due to the reference manager used. I would advise doing some of the changes manually to ensure correctness.

Please check the following:

References 1-4, 8, 9,18

Please check reference 1 (Organization WH) to World Health Organization.

We look forward to receiving your revised manuscript.

Kind regards,

Joseph Olusesan Fadare

Academic Editor

PLOS ONE

Journal Requirements:

Additional Editor Comments:

Thank you for updating the manuscript.

However, there are some outstanding issues. This may be due to the reference manager used. I would advise doing some of the changes manually to ensure correctness.

Please check the following:

References 1-4, 8, 9,18

Please check reference 1 (Organization WH) to World Health Organization.

---

## [Author Response · Author response to Decision Letter 2]

11 Oct 2023

We thank you for identifying references which were missing the journal names. We have reviewed the reference list to ensure correctness (Lines 289-342). Reference name Organization WH has been corrected to World Health Organization. References labelled 1-4, 8, 9 and 18 now have journal names (1& 2: The Lancet Infectious Diseases; Reference 4: Plos One; Reference 8: Malaria Journal; Reference 9: Saudi Pharmaceutical Journal; Reference 18: Antibiotics)

---

## [Editor Report · Decision Letter 3]

17 Oct 2023

Practices and Drivers of Self-Medication with Antibiotics among Undergraduate Medical Students in Eastern Uganda: A Cross-sectional Study

PONE-D-23-21270R3

Dear Mr Epuitai,

We’re pleased to inform you that your manuscript has been judged scientifically suitable for publication and will be formally accepted for publication once it meets all outstanding technical requirements.

Kind regards,

Joseph Olusesan Fadare

Academic Editor

PLOS ONE

Additional Editor Comments (optional):

Thank you for addressing the issues raised.
---

## [Editor Report · Acceptance letter]

11 Dec 2023

PONE-D-23-21270R3 

Practices and Drivers of Self-Medication with Antibiotics among Undergraduate Medical Students in Eastern Uganda: A Cross-sectional Study. 

Dear Dr. Epuitai:

I'm pleased to inform you that your manuscript has been deemed suitable for publication in PLOS ONE. Congratulations! Your manuscript is now with our production department. 

Kind regards, 

on behalf of

Dr. Joseph Olusesan Fadare 

Academic Editor

PLOS ONE